# A Numerical Study on Travel Time Based Hydraulic Tomography Using the SIRT Algorithm with Cimmino Iteration

**Pengxiang Qiu [1],[†]** , **Rui Hu [2],[†]** , **Linwei Hu [3]** , **Quan Liu [1]** , **Yixuan Xing [1]** , **Huichen Yang [1]** , **Junjie Qi [2] and Thomas Ptak [1],\***

[1] Applied Geology, Geoscience Centre, University of Goettingen, Goldschmidt Str. 3, 37077 Goettingen, Germany; pqiu@uni-goettingen.de (P.Q.); quan.liu@geo.uni-goettingen.de (Q.L.); yixuan.xing@geo.uni-goettingen.de (Y.X.); huichen.yang@stud.uni-goettingen.de (H.Y.)

[2] School of Earth Science and Engineering, Hohai University, Focheng Xi Road 8, Nanjing 211100, China; rhu@hhu.edu.cn (R.H.); jjqi@hhu.edu.cn (J.Q.)

[3] Institute of Geosciences, University of Kiel, Ludewig-Meyn-Straße 10, 24015 Kiel, Germany; linwei.hu@ifg.uni-kiel.de

\* Correspondence: tptak@gwdg.de; Tel.: +49-551-3913667

† First author.

**Abstract:** Travel time based hydraulic tomography is a technique for reconstructing the spatial distribution of aquifer hydraulic properties (e.g., hydraulic diffusivity). Simultaneous Iterative Reconstruction Technique (SIRT) is a widely used algorithm for travel time related inversions. Due to the drawbacks of SIRT implementation in practice, a modified SIRT with Cimmino iteration (SIRT-Cimmino) is proposed in this study. The incremental correction is adjusted, and an iteration-dependent relaxation parameter is introduced. These two modifications enable an appropriate speed of convergence, and the stability of the inversion process. Furthermore, a new result selection rule is suggested to determine the optimal iteration step and its corresponding result. SIRT-Cimmino and SIRT are implemented and verified by using two numerical aquifer models with different predefined ("true") diffusivity distributions, where high diffusivity zones are embedded in a homogenous low diffusivity field. Visual comparison of the reconstructions shows that the reconstruction based on SIRT-Cimmino demonstrates the aquifer's hydraulic features better than the conventional SIRT algorithm. Root mean square errors and correlation coefficients are also used to quantitatively evaluate the performance of the inversion. The reconstructions based on SIRT-Cimmino are found to preserve the connectivity of the high diffusivity zones and to provide a higher structural similarity to the "true" distribution.

**Keywords:** hydraulic tomography; travel time inversion; inversion algorithm; inverse problem

---

## 1. Introduction

Tomography is a technique for imaging sections of objects by using penetrating waves. The variation of wave signals between a transmitter and a detector can be analyzed and utilized to reconstruct the distribution of relevant parameters within the investigated object. Since the 1970s, Computed Tomography (CT) has become an important medical tool [1]. During a CT experiment, X-rays are absorbed to varying degrees when passing through different human body parts, and the attenuation of the X-ray radiation is measured and used to image the scanned body part. Seismic tomography is another application of the tomographical principle based on travel time. Seismic waves (P-waves, S-waves and surface waves) travel through different geological media at different velocities.

Their travel times are observed and recorded to reconstruct the subsurface structure with respect to the distribution of the velocity field [2]. A notable difference between these two applications is that the X-ray radiation travels through a human body along a straight line, while seismic waves are reflected and refracted in tectonic structures [3].

Two reconstruction algorithms are widely used in tomography: Algebraic Reconstruction Technique (ART) and Simultaneous Iterative Reconstruction Technique (SIRT). Both algorithms are common implementations of the Kaczmarz algorithm [4]. ART updates the reconstruction after analysis of a single travel time, while SIRT updates the reconstruction after analysis of the whole set of travel times. Compared to ART, SIRT has higher computational stability and is less sensitive to initial values and measurement errors [5–7]. Its main disadvantage is higher computational cost. However, considering the rapid development of computational hardware, the relevance of this disadvantage has been greatly reduced.

Over the past two decades, hydraulic tomography has been developed to determine the spatial distribution of aquifer hydraulic parameters [8–30]. Performance of a series of hydraulic tests (e.g., pumping tests and slug tests) with a tomographical configuration is the first step to gather the required hydraulic response data of the aquifer. During each test, a certain interval of a test well is isolated (e.g., with a packer system). This system allows water to be pumped out or injected into the aquifer only through this predefined section (source). The observation interval in other wells is also equipped with packer systems (or multi-chamber) to record the hydraulic response (e.g., the groundwater head) at different locations and depths (receivers). As the location of the source and receivers vary, a large number of response data can be collected. With appropriate inversion algorithms, the spatial distribution of the aquifer's hydraulic properties can be reconstructed, and the non-uniqueness and uncertainty of the inversion can be reduced.

Two main hydraulic tomography strategies have been developed over the years. The first strategy is based on solving the groundwater flow equation with numerical modeling. With a numerical model, calculated values are fitted to the observed (measured in-situ) values by optimizing input parameters of the model (parameter estimation) [28,31–34]. With this strategy, the distribution of different kinds of hydraulic parameters can be derived at a high resolution. However, the large amount of field tests required, and the parameter estimation process itself, are time consuming.

The second strategy is Travel Time based Hydraulic Tomography (TTHT), which applies an asymptotic approach to transform the groundwater flow equation into the eikonal equation. Ray-tracing techniques can then be utilized to describe the transient pressure propagation and solve the eikonal equation. With this strategy, a line integral is derived, which relates the travel time of the transient pressure signals to a hydraulic diffusivity distribution [30,35–41]. In seismic tomography, the travel time of waves between a source and a receiver have a line integral relationship with the velocity field within the structure. This is similar to CT, where the decimal percent drop in X-ray intensity is linearly related to the attenuation as a line integral [42]. These similarities imply the feasibility of using SIRT and ART algorithms for hydraulic studies [37,41]. Based on similar principles, another strategy is further developed to obtain the spatial distribution of the specific storage using hydraulic attenuation inversion [38]. Compared to the above introduced strategy, the hydraulic travel time based tomography can only solve the spatial distribution of hydraulic diffusivity and specific storage.

The advantages of hydraulic travel time based tomography (following the one used for seismic tomography) are high computational efficiency and robustness. Huge data sets with thousands of travel times can be handled within minutes on a common computer. Following the principles of seismic tomography, a line integral can be derived relating the square root of the pressure response arrival time directly to the square root of the reciprocal of diffusivity. In this study, the similarity between the hydraulic line integral and seismic line integral will be exploited by using the same inversion techniques. Since the arrival of hydraulic travel time signal is so early (within minutes, even seconds), that the boundary effect of the aquifer has not even taken place, a further advantage of the travel time integral is that a change in boundary conditions during a test (e.g., nearby pumping, recharge) barely

influences the travel time propagation of a pressure response from a "defined source," and thus the inversion result is not affected.

In the iterative reconstruction algorithm, a residual represents the difference between the calculated travel time and the observed travel time. This value reflects the convergence of the algorithm towards a possible solution. Related studies have shown that even though the residual may be already convergent, the reconstruction results are highly dependent on the number of iterations, and the applied number of iterations is usually determined (or given) empirically [38]. To the best of our knowledge, a reliable result selection rule has not yet been developed for TTHT.

This work focuses on the further utilization of SIRT in TTHT. Since the relationship between travel time and hydraulic diffusivity is described as a line integral, it can be discretized and transformed into a matrix equation. The SIRT is then executed step by step to solve this matrix equation. Considering the drawbacks of SIRT, the Cimmino iterative method is introduced and embedded within SIRT (SIRT-Cimmino) in this study. A new result selection rule for SIRT-Cimmino is proposed to determine the optimal iteration step and its corresponding result. For the validation of this method, two numerical aquifer models with a predefined diffusivity distribution ("truth") are designed. The first model represents an inclined stratified aquifer, while the other model represents a more complex lying Y-shaped aquifer. Pumping tests are simulated with these two models, and the calculated groundwater heads (observations) are analyzed to obtain hydraulic travel times. Finally, SIRT and SIRT-Cimmino are implemented, and their performance under different spatial resolutions are evaluated by Root Mean Square Errors (RMSE) and correlation coefficients, based on the comparison between the inversion results and the "truth".

## 2. Methodology

The start of a pumping process can be considered as a generation of a Heaviside signal. This signal is released at the pumping location, propagates within a saturated aquifer, and arrives at the observation location. Travel time $t_{peak}$ is defined as the time at which the pressure pulse (the first derivative of drawdown with respect to time) reaches its maximum amplitude, and the relationship between $t_{peak}$ and diffusivity can be described as [37,41,43]:

$$\sqrt{t_{peak}(x_2)} = \frac{1}{\sqrt{c}} \int_{x_1}^{x_2} \frac{ds}{\sqrt{D(s)}}$$

(1)

where $t_{peak}(x_2)$ is the travel time of the signal from a releasing point $x_1$ (source) to an observation point $x_2$ (receiver), $s$ is the propagation path of the signal, $D(s)$ is the diffusivity along the path, and $c$ is a dimension dependent coefficient. $c$ is 4 for a 2D case and 6 for 3D cases [41,43].

The concept of different travel times is generalized through the introduction of travel time diagnostic [14,36,37]. As an example, the $t_{10}$ diagnostic is the time when the first derivative of drawdown rises to 10% of its maximum amplitude. According to this definition, the $t_{peak}$ is defined as $t_{100}$. Figure 1 shows three kind of travel times: $t_{10}$, $t_{50}$, and $t_{100}$. Since our study focuses on the performance of the reconstruction algorithm, only $t_{100}$ is utilized for the inversion.

A transformation factor $f$ is introduced by Brauchler et al. [37], and the relationship between travel time diagnostics and diffusivity is described as follows:

$$\sqrt{t} = \frac{1}{\sqrt{cf}} \int_{x_1}^{x_2} \frac{ds}{\sqrt{D(s)}}$$

(2)

where $t$ is the travel time diagnostic, $f$ is the transformation factor which is a travel time diagnostic related Lambert's W function and which can be determined numerically. In this study only $t_{100}$ was utilized. For more information about the transformation factor, the reader is referred to the work performed by Brauchler et al. [37] and Hu [38].

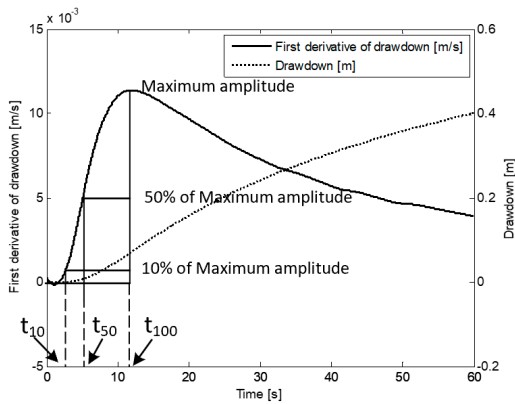

**Figure 1.** Three kinds of travel time $t_{10}, t_{50}$, and $t_{100}$ [14,37].

## 2.1. Discretization of the Line Integral

The investigation domain ($\Omega$) between a pumping well and an observation well is divided into a grid of $n$ small rectangular cells ($\Omega_1, \cdots, \Omega_n$). $D_j$ is the mean diffusivity value in cell $\Omega_j$. In this study, the cells were distributed as a matrix, and the distribution resolution is defined as the number of rows multiplied by the number of columns. Therefore, the resolution is $4 \times 6 = 24$ and $n = 24$ in Figure 2. The pressure signal propagates along the shown curve from the source through $\Omega$ to the receiver.

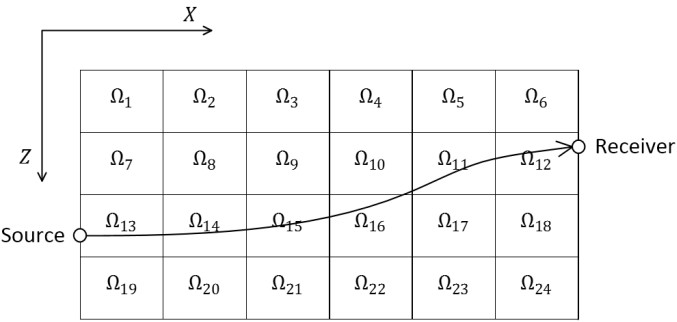

**Figure 2.** Signal travels through a discrete area.

By providing the investigation domain with more sources and receivers, more pressure signals can be sampled. Assuming that $m$ signals are given, $s_{ij}$ is the length of $i$th signal trajectory in $\Omega_j$. $s_{ij} = 0$ if this $i$th trajectory does not traverse $\Omega_j$.

Equation (1) is thus rewritten as:

$$\sqrt{ct_{100}} = \sum_{j=1}^{n} \sqrt{\frac{1}{D_j}} s_{ij}, i = 1, \cdots, m. \tag{3}$$

In matrix form, Equation (3) becomes:

$$b = Ax$$

$$b_i = \sqrt{ct_{100}}, A_{ij} = s_{ij}, x_j = \sqrt{\frac{1}{D_j}} \tag{4}$$

$$i = 1, \cdots, m, j = 1, \cdots, n,$$

where $b$ is an $m$-vector consisting of $m$ observed travel times, and $A$ is an $m \times n$ matrix. The matrix element represents the trajectory length in each cell, and each row represents a signal propagation. In other words, $A$ records the propagation path of all signals in the investigation domain. $x$ is an

*n*-vector, each vector element is related to the diffusivity of each cell, and the vector *x* shows the diffusivity distribution of the investigation domain.

The above introduced inversion requires solving the linear-like matrix Equation (4). However, use of this equation does not suggest a linear relationship. According to the Fermat Principle, a pressure signal propagates from one point to another point along the path that takes the least travel time [44]. In our case, signals preferentially penetrate cells with high diffusivity, since higher diffusivity leads to less travel time according to Equation (1). Thus, the propagation trajectory is related to the diffusivity distribution. In other words, *A* is derived from *x* in Equation (4). Matrix Equation (4) should therefore be categorized as a nonlinear problem [42].

### 2.2. SIRT and SIRT-Cimmino Algorithms

Traditionally, when applying the SIRT algorithm to solve the nonlinear problem, the residual has been used to determine the convergence. However, even with the same degree of convergence, results with a different number of iteration steps (NIS) show large differences. Hence, a clearly defined selection rule is required for the determination of NIS. To overcome this drawback of SIRT, a Cimmino iterative method based SIRT algorithm is proposed.

Given an *n*-vector *x*, the Euclidean norm of *x* is denoted as $\|x\| = \sqrt{x^T x}$ and the *M*-norm of *x* is denoted as $\|x\|_M = \sqrt{x^T M x}$, where *M* is an *n* x *n*-matrix. In this study, when discussing an iterative algorithm, the superscript with parenthesis stands for the number of iterations and the subscript stand for the position of element, e.g., $x_i^{(k)}$ is the *i*th element of *x* in the *k*th iteration, and $A_{ij}^{(k)}$ is the element in the *i*th row and the *j*th column of *A* in the *k*th iteration.

The SIRT algorithm for travel time based hydraulic tomography can be introduced with the following steps:

Step 1. Initialization and criteria. If prior information about diffusivity distribution is lacking, a homogeneous diffusivity distribution with an initial value ($x^{init}$) is suggested, with the assumption that all signals travel along straight lines through the domain of interest. Two criteria are feasible in this algorithm: either a fixed residual-dependent tolerance or a fixed number of iterations. The algorithm performance at different iteration numbers is the focus of this study, so the second criterion is chosen.

Step 2. Utilization of ray-tracing technique. Ray-tracing is a tool to determine the path of signal propagation. This process reconstructs $A^{(k)}$ based on $x^{(k)}$.

Step 3. Residual ($\Delta b^{(k)}$) calculation with

$$\Delta b^{(k)} = b - b^{(k)} = b - A^{(k)} x^{(k)} \tag{5}$$

Step 4a. Incremental correction. To approach the solution, an incremental correction $\Delta x^{(k)}$ is built into each iteration with:

$$\Delta x^{(k)} = W^{(k)} A^{(k)T} N^{(k)} \Delta b^{(k)} \tag{6}$$

where the diagonal matrix $W^{(k)} = diag\left(\frac{1}{w_1}, \cdots, \frac{1}{w_n}\right)$, and $w_j$ is the number of nonzero values in the *j*th column of $A^{(k)}$; the diagonal matrix $N^{(k)} = diag\left(\frac{1}{\|a_1\|^2}, \cdots, \frac{1}{\|a_m\|^2}\right)$, with $a_i$ representing the *i*th row vector of $A^{(k)}$ and $\|a_i\|$ representing the Euclidean norm of $a_i$. Both $W^{(k)}$ and $N^{(k)}$ are derived from $A^{(k)}$ depending on the iteration step *k*. Hence, $W^{(k)}$ and $N^{(k)}$ are iteration-dependent.

Step 5. Iteration updates. The diffusivity vector *x* is updated based on $x^{(k)}$ and incremental correction $\Delta x^{(k)}$ with:

$$x^{(k+1)} = x^{(k)} + \Delta x^{(k)} \tag{7}$$

The algorithm stops if $k + 1$ reaches the iteration step set as a criterion in Step 1. Otherwise the calculation continues to the next iteration.

In our application, matrix $A$ is rebuilt every iteration with minimal influence on the algorithm convergence. Due to the specific iterative form in Equation (6), two drawbacks could cause inversion failure: (a) if there is a cell $\Omega_j$ that is not traversed by any ray, then the $j$th column of $A$ is a zero vector, $w_j = 0$, and $W^{(k)} = diag\left(\frac{1}{w_1}, \cdots, \frac{1}{w_n}\right)$ would not be generated since the denominator of $\frac{1}{w_j}$ is zero; and (b) the result (diffusivity distribution) at one iteration step differs from the results at other iterations. In practice, the number of steps cannot be determined only by the residual convergence. Studies have shown that fewer iterations with a higher residual may possibly lead to better solutions and more iterations with a lower residual may lead to higher deviation [10]. Thus, a reliable and feasible rule to determine the optimal number of iteration steps and the selection of the best reconstruction is needed.

In order to overcome these drawbacks, a Cimmino iterative method based SIRT algorithm (SIRT-Cimmino) is proposed (Figure 3) [45,46]. The Cimmino iterative method can modify the incremental correction to avoid the case of division by zero (Equation (8)). An iterative dependent relaxation parameter is introduced to adjust the convergence velocity and a result selection rule for SIRT-Cimmino is suggested. In Chapter 4.2, this rule is presented, and its feasibility will be proven with numerical tests. Step 4a in SIRT is modified by following Step 4b.

Step 4b. Incremental correction,

$$\Delta x^{(k)} = \lambda_k A^{(k)T} M^{(k)} \Delta b^{(k)} \tag{8}$$

where matrix $M^{(k)} = \frac{1}{m} diag\left(\frac{1}{\|a_1\|^2}, \cdots, \frac{1}{\|a_m\|^2}\right)$, from which $a_i$ is the $i$th row vector in $A^{(k)}$, $m$ is the number of rows in $A^{(k)}$, and $\lambda_k$ is the relaxation parameter defined below,

$$\lambda_k = \frac{\Delta b^{(k)T} M^{(k)} \Delta b^{(k)}}{\|A^{(k)T} M^{(k)} \Delta b^{(k)}\|_2^2} \tag{9}$$

The relaxation parameter in the iteration algorithm can be either a constant (iteration-independent) or a variable (iteration-dependent). In CT, X-rays traverse human tissue and organs along straight lines, which implies that $A$ is a fixed matrix and iteration-independent. Thus, Equation (4) becomes a linear problem and the iteration steps of SIRT-Cimmino converge to a solution $x^*$ of $min\|Ax - b\|_M$, if $\lambda \in \left(0, \frac{2}{\sigma^2}\right)$, where $\sigma$ denotes the maximum singular value of $A^T MA$ [47]. The Matrix $A$ is rebuilt in each iteration, and varies while the diffusivity field is regenerated by incremental correction. Therefore, it is reasonable to define an $A$-dependent $\lambda$ considering convergence. In 1987, Dos Santos [48] proposed the construction of $\lambda$ in Equation (8), which minimized $\|x^* - x^k\|$ in each iteration, where $x^*$ is a solution of $Ax = b$ [47].

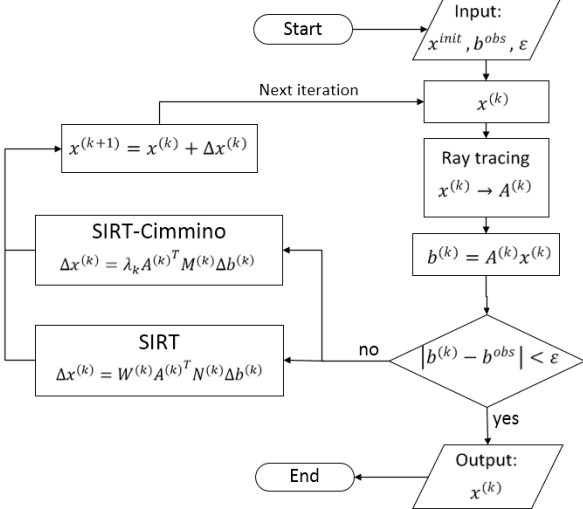

**Figure 3.** Flowchart of SIRT (Simultaneous Iterative Reconstruction Technique) and SIRT-Cimmino algorithms.

### 2.3. Ray-Tracing Technique

Ray-tracing is a method to describe the signal trajectory numerically and has three approaches: straight rays, ray bending, and network theory with minimum-time paths. In the first approach, every trajectory is approximated as a straight line. This approach is the fastest but is generally the least accurate, especially when the aquifer is heterogeneous. The ray bending approach adjusts ray paths by calculating local gradients. This method is fast and more accurate, but may lead to a ray path with a local minimum time instead of a global minimum time. Network theory is based on a fixed grid of nodes. Every node can link to other nodes nearby through straight line segments. The trajectory from one node to another node is approximated using the connection of these line segments. All possible connections between source and receiver are considered and the trajectory with the minimum travel time will be chosen. This method takes the longest calculation time but guarantees a global minimum time [49]. Since it is more accurate, our experiments apply this theory to find the ray path of a hydraulic pressure signal.

Figures 4 and 5 show a grid that was proposed by Moser [44]. Figure 4 presents the distribution of nodes in one cell. Each cell has two nodes on every edge. Each node can only connect with other nodes that are in the same cell and are not on the same edge. As Figure 5 shows, the cells spanned by line segments form a network, so that a path from S to R can be approximated. The Dijkstra algorithm is widely used to find the shortest path in a network. By replacing distance with travel time, the Dijkstra algorithm can be used for our problem. Further information about this algorithm is available from Dijkstra [50] and Nakanishi and Yamaguchi [51]. Note that the accuracy of the network theory is strongly dependent on the density of nodes in a single cell and the density of cells in a grid (the reconstruction resolution). Accuracy can be improved by increasing either of the two densities, however at the cost of a more time-consuming computation. Hence, a small travel time path may not even be uniquely determined. For example, assuming the investigation area in Figure 5 is homogeneous, and every line segment has equal weight, the orange path has the same travel time as the blue path. This problem would occur more often in heterogeneous cases, since each line segment could have a different weight.

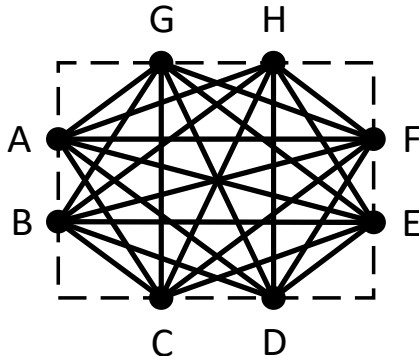

**Figure 4.** Node distribution in a cell.

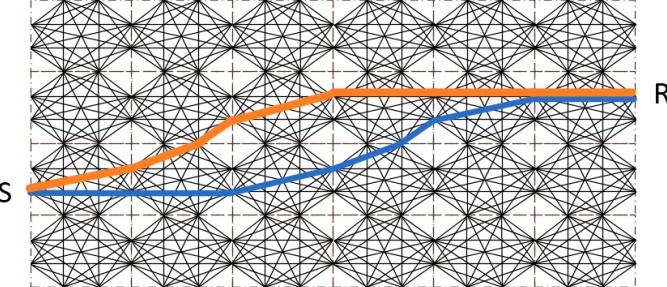

**Figure 5.** Two possible paths (orange and blue) from S to R in a network.

### 2.4. Residual, Root-Mean-Square Error and Correlation Coefficient

The residual describes the difference between observed travel times and the approximated travel times from ray-tracing in each iteration, and is given by:

$$R = \frac{\sqrt{\sum_{i=1}^{m}\left(\sqrt{t_i^{(k)}} - \sqrt{t_i}\right)^2}}{\sum \sqrt{t_i}} \tag{10}$$

where $t_i^{(k)}$ denotes the approximated travel time of the *i*th signal in the *k*th iteration, and $t_i$ denotes the observed travel times of the *i*th signal (in our case forwardly calculated through the numerical model) [14,37].

In this work, the predefined "truth" and the estimated diffusivity distributions are studied in 2D. Therefore, two measurements for the comparison are introduced: Root Mean Square Error (RMSE) and correlation coefficient (*C*). RMSE estimates the difference between two distribution images on a "pixel-by-pixel" basis [52,53] and is defined by:

$$RMSE = \sqrt{\frac{\sum_{i=1}^{n}\left(D_i^{est} - D_i\right)^2}{n}} \tag{11}$$

where $D_i^{est}$ and $D_i$ denote the estimated and original diffusivity in the *i*th cell, respectively.

*C* measures the degree of correlation between two distribution images and is given by:

$$C = \frac{\sum_i^n\left(D_i^{est} - \overline{D^{est}}\right)\left(D_i - \overline{D}\right)}{\sqrt{\left(\sum_i^n\left(D_i^{est} - \overline{D^{est}}\right)^2\right)\left(\sum_i^n\left(D_i - \overline{D}\right)^2\right)}} \tag{12}$$

where the notation corresponds to Equation (11), and

$$\overline{D^{est}} = \frac{1}{n}\sum_{i=1}^{n}D_i^{est}, \overline{D} = \frac{1}{n}\sum_{i=1}^{n}D_i \tag{13}$$

$\overline{D^{est}}$ and $\overline{D}$ denote the average of the estimated and original diffusivity in the entire research area, respectively.

In our case, the correlation coefficient describes the similarity between two distribution images [53], and ranges from −1 to 1. (*C* = 1 if the two images are completely identical, *C* = 0 if they are uncorrelated, and *C* = −1 if they are anti-correlated.)

## 3. Numerical Study

Two 2D axisymmetric numerical groundwater flow models (models A and B) are built using the finite element software COMSOL Multiphysics® to simulate a series of short-term pumping tests with transient flow conditions. Each model consists of three parts: the area of interest for parameter estimation (4 m × 2.8 m, blue zone in Figure 6), the homogeneous zone (10 m × 2.8 m, dark grey in Figure 6), and the homogeneous surrounding domain (24 m × 30 m, light grey in Figure 6).

The two models differ with respect to the parameter distribution within the area of interest. Model A is provided with an inclined high-diffusivity (high-D) band (Figure 7a), while model B has a lying Y-shaped high-diffusivity zone (Figure 7b). The remaining background low-diffusivity (low-D) zone is homogenous. Regarding the possible range of hydraulic parameters of the fluvio-sedimentary aquifer, the diffusivity value is set to 10 m²/s within the high-D area (green band in Figure 7) and 0.2 m²/s within the low-D zone (blue zone in Figure 7), based on earlier studies [14,54]. Therefore, the diffusivity contrast ratio is 50.

To eliminate the influence of boundary effects during the simulation, an infinite element domain with a scaling factor of 1000 was added to the right of the study zone with constant head (Figure 6, with only the first 30 m shown). In this domain, rational coordinate scaling was utilized to stretch the finite element domain where the dependent variables vary less with radial distance. The surrounding area was designated as a homogenous isotropic material, to maintain the continuity of diffusivity at the boundary of the area of interest. According to the measured value of hydraulic parameters in a fluvio-sedimentary aquifer by Hu [38], the hydraulic conductivity, specific storage and porosity were set as $K = 8 \times 10^{-5}$ m/s, $S_s = 4 \times 10^{-4}$ 1/m, and 0.2 (−), respectively. The diffusivity $D$ was therefore set as 0.2 m²/s, since $D = \frac{K}{S_s}$.

Each simulated pumping test had a pumping duration of ten minutes, and the head sampling interval was 0.02 s. The area of interest size and the pumping and observation positions are shown in Figure 7. Eight pumping and eight observation positions are represented by S1, ... ,S8 and R1, ... ,R8, respectively. The radius of the pumping well and observation well was 0.05 m. The initial head of the aquifer was set to 10 m. The pumping tests were simulated sequentially at each pumping position while the head changes are recorded at all eight observation points throughout each test. This resulted in 8 × 8 drawdown data sets. As an example, Figure 8a shows the head data recorded at the eight observation points when the pumping test was performed at S1. During each simulated pumping test, the constant head boundary was not reached.

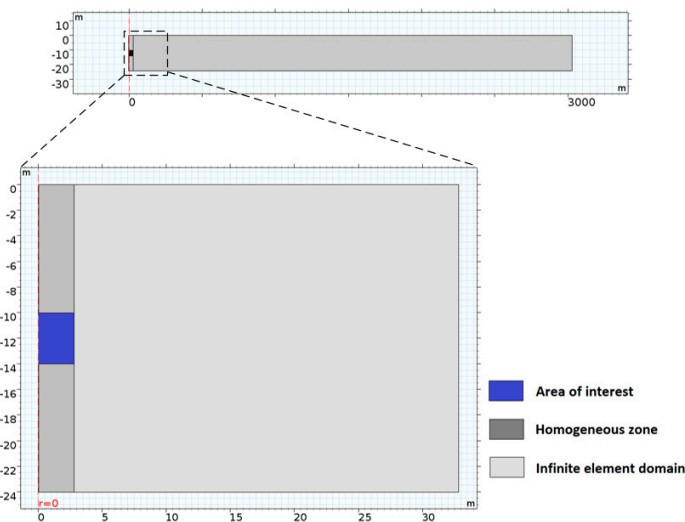

**Figure 6.** The geometry of the 2D axisymmetric model.

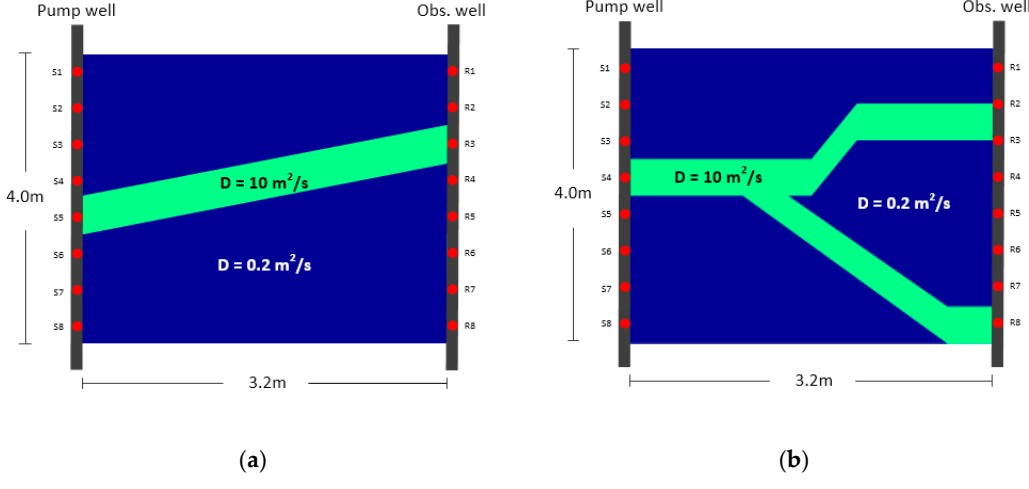

**(a)**                    **(b)**

**Figure 7.** Predefined diffusivity distribution (**a**) Model A, and (**b**) Model B.

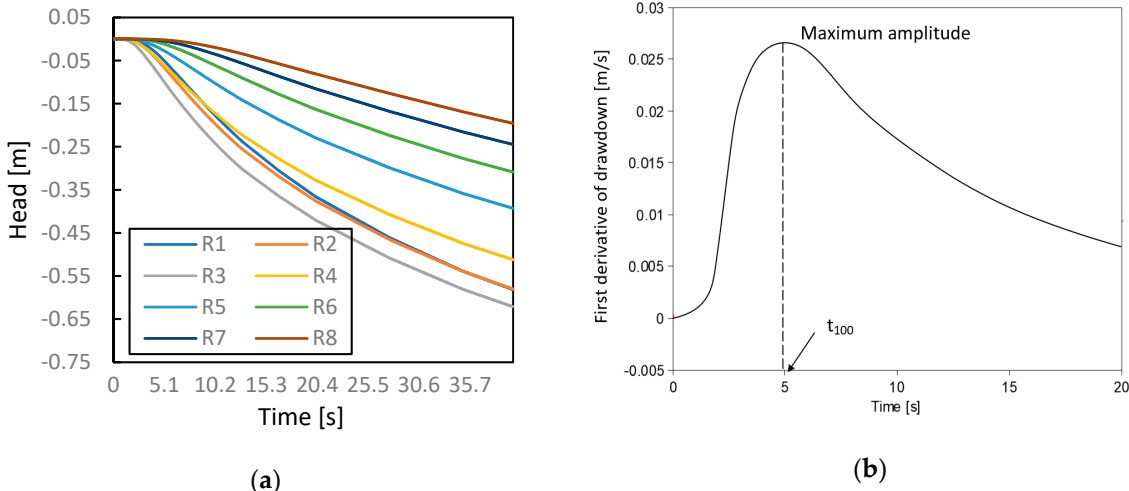

**Figure 8.** (**a**) Head drawdown recorded at R1–R8 while pumping at S1, (**b**) first derivative of drawdown recorded at R3, and the correspondent $t_{100}$ while pumping at S1.

The pressure signal travel time was derived by applying first order differentiation to the recorded drawdown curves. In Figure 8b, three types of travel time diagnostics are shown. Since our study focuses on the performance of the reconstruction algorithm, only $t_{100}$ was utilized for the inversion.

The inversion was implemented with programming language C#, and performed using a laptop computer with an Intel(R) Core(TM) i7 CPU Q840 under the Windows 7 operating system.

In our inversion process, in addition to the inversion algorithm (SIRT and SIRT-Cimmino), it is important to consider the resolution of the inversion result and the ray-tracing technique employed.

The resolution mainly depends on the number of signals used for the inversion. High inversion resolution with less non-uniqueness and uncertainty is the aim of this study. Three kinds of resolutions were considered in our experiments.

## 4. Results

### 4.1. Result Selection Rule for SIRT

The number of iteration steps (NIS) is a key parameter for the reconstruction process. Therefore, a selection rule to determine an appropriate NIS must be defined and verified.

As NIS increases, the travel time residual in SIRT is reduced, and converges to a non-zero constant (Figure 9). Due to the difficulty of non-linearity, a standard NIS cannot be determined mathematically. Brauchler et al. [37] empirically utilized an NIS of 10, and Hu [38] utilized an NIS of 8.

For the inversion constraint, we first assume that the investigation domain is homogeneous, and every trajectory in this domain is a straight line. By using the above mentioned straight ray inversion approach as the first inversion step, a vector with uniform diffusivity value is obtained. This diffusivity value is set as the average value of the elements in the vector, i.e., the initial value of diffusivity for the heterogeneous domain in the following inversion steps. In our case, a range for diffusivity during the inversion calculation was set with lower and upper limits defined as 0.01 times and 100 times the initial diffusivity, respectively. In Model A, the initial diffusivity was set as 0.78 $m^2/s$, the lower and upper limits were 0.0078 $m^2/s$ and 78 $m^2/s$, respectively.

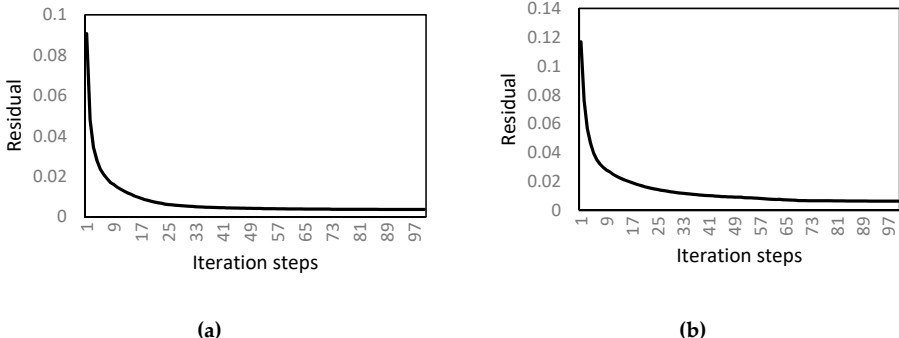

**Figure 9.** Residual for 100 iteration steps by using SIRT under 8 x 8 resolution in different models, (**a**) Model A, and (**b**) Model B.

The inversion results at the 10th, 25th, and 100th steps are shown in Figure 10b–d. Each of these three distributions indicates the existence of a high-D zone connecting S5 and R3. Comparison with the predefined distribution (Figure 10a) shows that the diffusivity values in the high-D zones are lower than 10. Diffusivity in the high-D zone increased as NIS increased, but were still too low in the center after 100 iterations, despite convergence of the residual after 25 iteration steps. After 25 steps, the diffusivity near S5 and R3 increased so rapidly that it reached the upper limit (78.49 m$^2$/s). This is in agreement with the findings by Brauchler et al. [37], who mentioned that a deviation may occur with a large NIS. Mathematically, the non-uniqueness is a possible explanation, and SIRT might converge to a wrong solution. After comparison between the inversion result at every iteration step (within 100 steps) and the "true" distribution in our case, the result at the 10th step was considered the best reconstruction, and therefore 10 was considered an appropriate value for NIS. Similarly, the optimal NIS for Model A and Model B (with different inversion resolutions) were determined (Table 1).

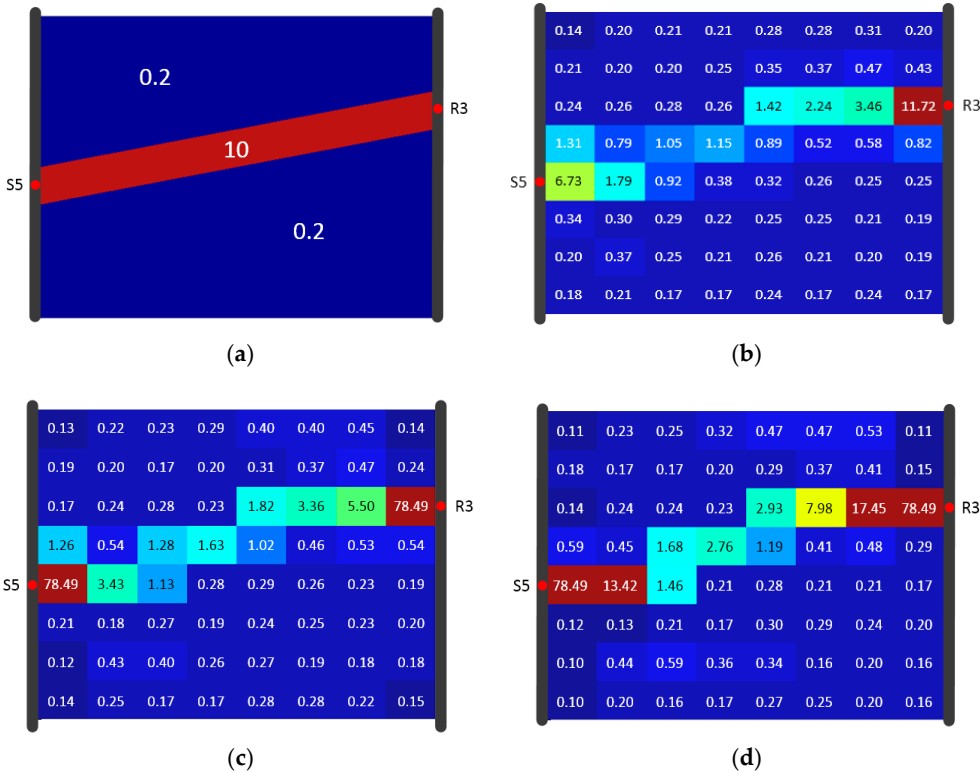

**Figure 10.** Diffusivity (m$^2$/s) tomograms based on the inversion of $t_{100}$ and SIRT under $8 \times 8$ resolution with different numbers of iterations. (**a**) Predefined diffusivity ("truth") distribution; and (**b–d**) inversion results after 10, 25, and 100 iterations, respectively.

**Table 1.** Optimal number of iteration steps for using SIRT under three different model resolutions.

|  | SIRT | | |
| --- | --- | --- | --- |
|  | **8 × 6** | **8 × 8** | **12 × 12** |
| Model A ($t_{100}$) | 11 | 10 | 5 |
| Model B ($t_{100}$) | 5 | 5 | 3 |

As shown in Table 1, the optimal NIS when using SIRT is dependent on the model and the model resolution. The influence of the travel time type (e.g., $t_{100}$, $t_{50}$ and $t_{10}$) was not investigated in this work. Of course, in practice, when the prior information on hydraulic parameters within the investigated area is insufficient, it would be hard for the SIRT user to determine the optimal NIS to obtain the best result.

*4.2. Result Selection Rule for SIRT-Cimmino*

Figure 11a,b show the residuals after 50 iteration steps when using SIRT-Cimmino in Model A and Model B, respectively. Oscillations were found in both models and the convergence was not easily determined. Mathematically, this behavior is explained by the rebuilding of matrix *A* at each iteration and the non-uniqueness of the solution. Rebuilding matrix *A* disturbs the residual convergence and even leads to a separate solution.

As shown in Figure 9, the residual is stabilized after several steps, as the algorithm trends toward a solution. This means the residual value belongs to a single solution. The divergent behaviors in Figure 11a,b therefore indicate several solution approaches, which are represented by subsequences. For instance, the green subsequence in each Figure indicates a possible solution approach. In Figure 11b, the oscillation ends after 40 steps, with the following residuals indicating another solution.

The selection of a result for SIRT-Cimmino is proposed through the following steps:

(1) Calculating 50 steps of iteration (due to computational time);
(2) Selecting a convergent subsequence with a low residual if convergent subsequences exist;
(3) Choosing the step with the lowest residual in this convergent subsequence as the optimal NIS and the corresponding result as the SIRT-Cimmino reconstruction.

In Figure 11a,b, the steps marked with black diamonds were chosen as the optimal iteration steps.

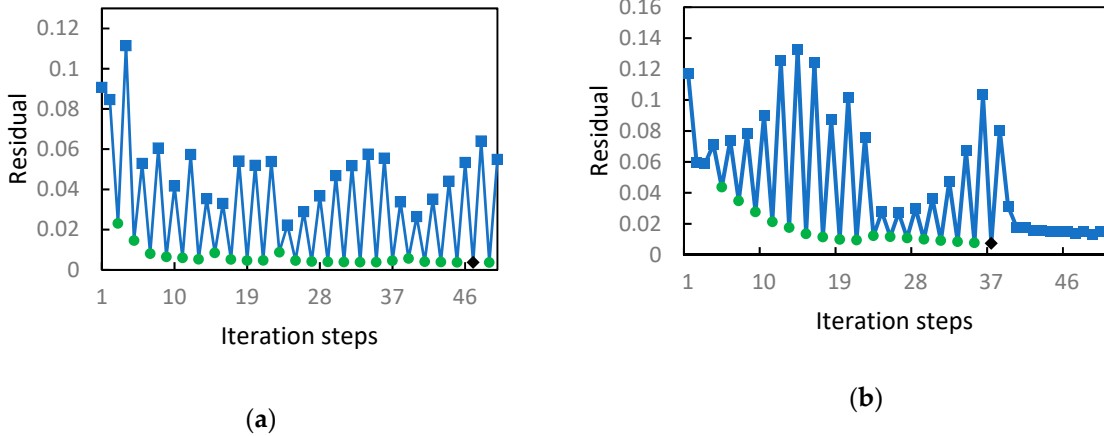

(**a**)

(**b**)

**Figure 11.** Residual of travel time for 50 iteration steps by using $t_{100}$ and SIRT-Cimmino under 8 × 8 resolution in (**a**) Model A and (**b**) Model B.

*4.3. Reconstruction comparison of SIRT and SIRT-Cimmino for Model A*

Both algorithms used $t_{100}$ data for reconstruction of the diffusivity distribution, given again in Figure 12. Figures 13–15 show the inversion results with resolutions of $8 \times 6$, $8 \times 8$, and $12 \times 12$ (using the same color scale). In each SIRT result (Figures 13a, 14a and 15a), the values in the high-D zones were nearly one. These values did not clearly distinguish the high-D zone from the background. In comparison, each SIRT-Cimmino result showed a clear high-D zone with better connectivity.

RMSE and the correlation coefficient were calculated and are listed in Table 2. The comparison showed that both algorithms have similar RMSE values. The SIRT-Cimmino had better performance with respect to the correlation coefficient. This means that the SIRT-Cimmino result delivers a higher similarity to the predefined distribution. In addition, the correlation coefficient increased as resolution increased, since the higher resolution improved the description of the high-D zone (the main structural feature). In other words, the correlation coefficient was extremely sensitive to this zone.

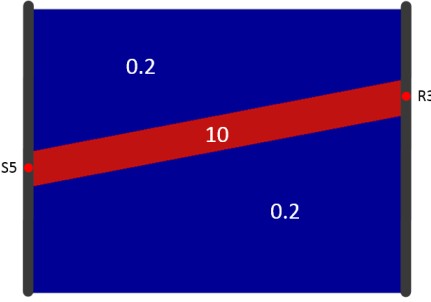

**Figure 12.** Predefined diffusivity ("truth") (m$^2$/s) distribution of Model A.

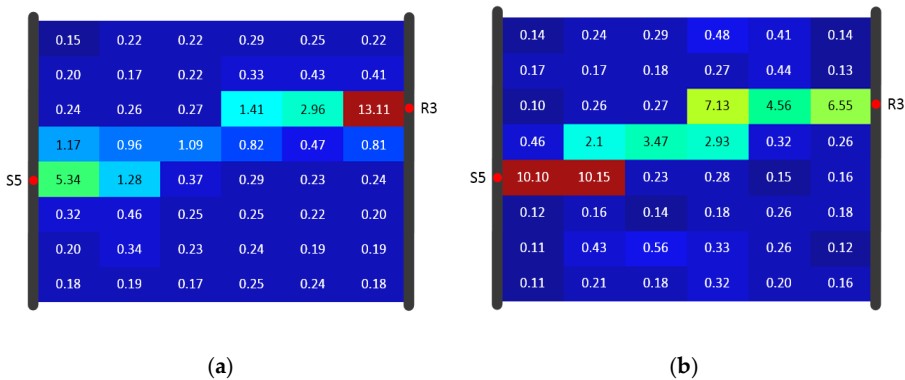

(**a**)          (**b**)

**Figure 13.** Algorithm result comparison for Model A under $8 \times 6$ resolution of the (**a**) SIRT result and (**b**) SIRT-Cimmino result, shown in diffusivity (m$^2$/s).

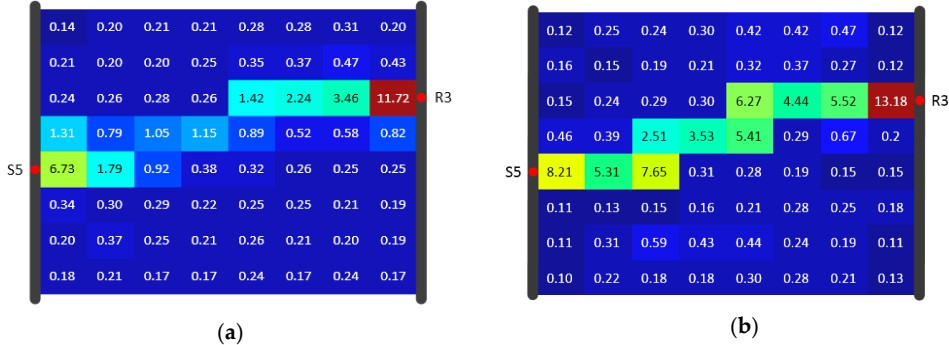

(**a**)          (**b**)

**Figure 14.** Algorithm result comparison for Model A under $8 \times 8$ resolution of the (**a**) SIRT result and (**b**) SIRT-Cimmino result, shown in diffusivity (m$^2$/s).

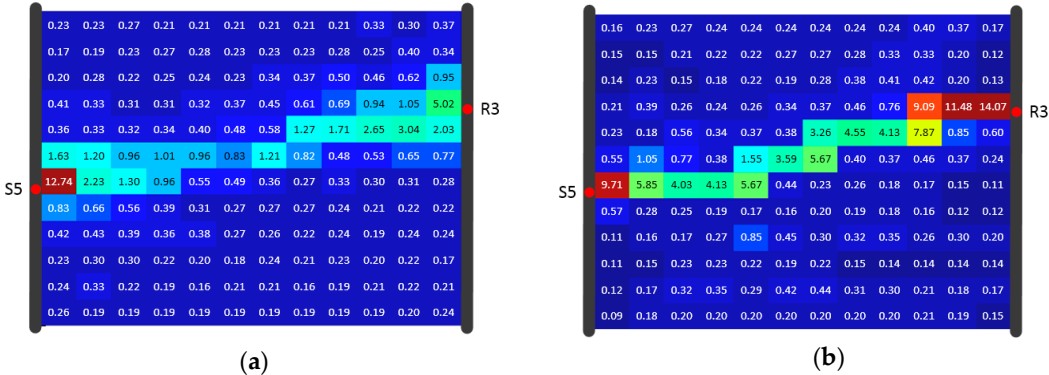

**Figure 15.** Algorithm result comparison for Model A under $12 \times 12$ resolution of the (**a**) SIRT result and (**b**) SIRT-Cimmino result, shown in diffusivity (m$^2$/s).

**Table 2.** RMSE (Root Mean Square Errors) and correlation coefficient of Model A inversion using SIRT and SIRT-Cimmino.

| | RMSE | | | Correlation Coefficient | | |
|---|---|---|---|---|---|---|
| | $8 \times 6$ | $8 \times 8$ | $12 \times 12$ | $8 \times 6$ | $8 \times 8$ | $12 \times 12$ |
| SIRT | 4.04 | 3.55 | 6.41 | 0.70 | 0.70 | 0.77 |
| SIRT-Cimmino | 2.86 | 3.77 | 4.24 | 0.73 | 0.72 | 0.79 |

*4.4. Reconstruction Comparison of SIRT and SIRT-Cimmino for Model B*

Both algorithms used $t_{100}$ data for reconstruction of the diffusivity distribution given in Figure 16. Figures 17–19 show the inversion results with resolutions of $8 \times 6$, $8 \times 8$, and $12 \times 12$ (using the same color scale), respectively. Comparison with the "true" diffusivity distribution (Figure 16) revealed that the lying Y-shaped high-D zone could be reconstructed. The reconstructions of SIRT were generally worse than the reconstructions from SIRT-Cimmino. This visual assessment coincided with the correlation coefficient calculation, since the correlation coefficient of SIRT-Cimmino was overall larger than that of SIRT, especially at the $12 \times 12$ resolution (Table 3).

The RMSE calculation did not show any advantage for SIRT-Cimmino in either Model A or Model B. There are two possible reasons. First, the discretization method assumes that the research area is divided into rectangles, which cannot approximate the inclined edge (shape) of the high-D zone perfectly. Second, the high values near R8 in SIRT-Cimmino influence the overall RMSE.

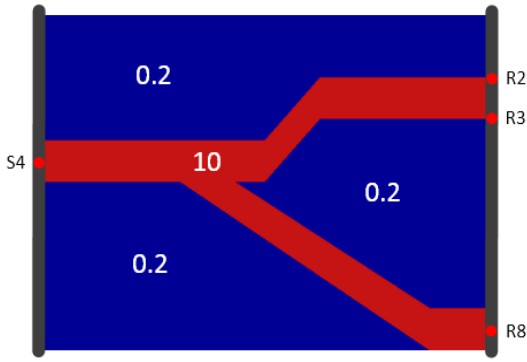

**Figure 16.** Predefined diffusivity distribution (m$^2$/s) of Model B.

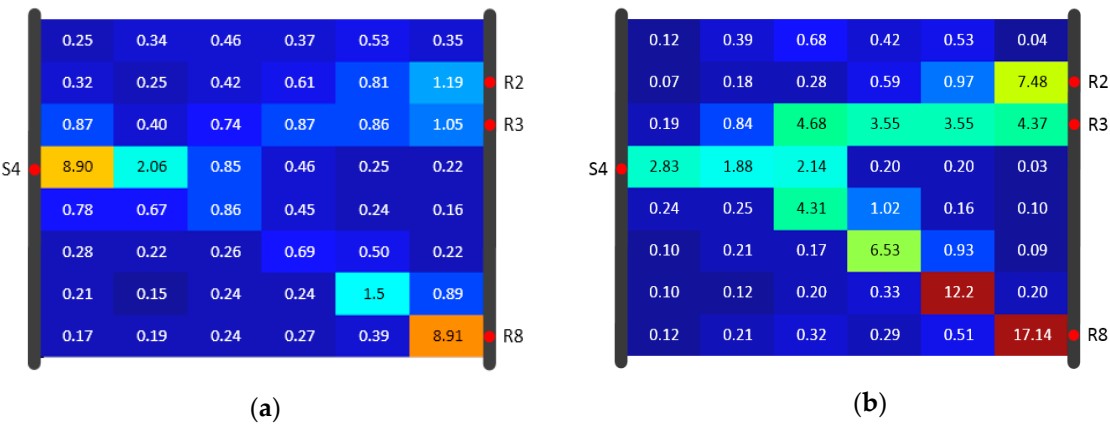

**Figure 17.** Algorithm result comparison for Model B under 8 × 6 resolution of the (**a**) SIRT result and (**b**) SIRT-Cimmino result, shown in diffusivity (m²/s).

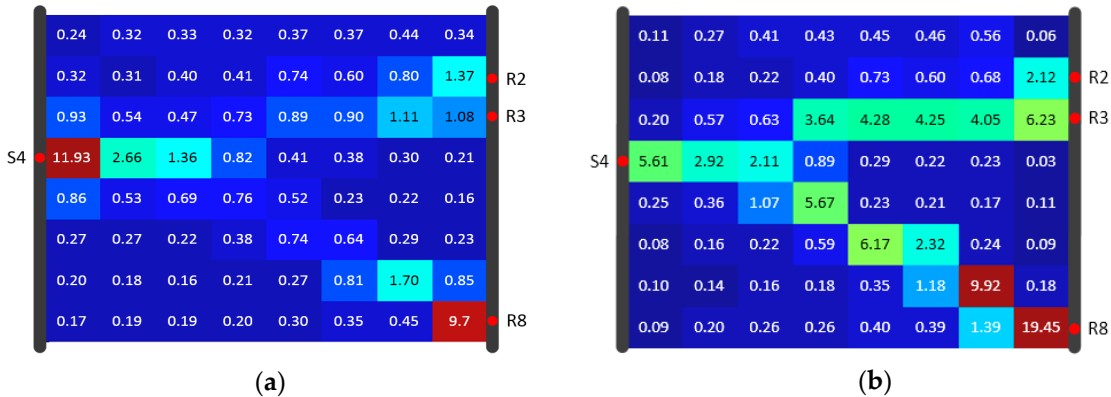

**Figure 18.** Algorithm result comparison for Model B under 8 × 8 resolution of the (**a**) SIRT result and (**b**) SIRT-Cimmino result, shown in diffusivity (m²/s).

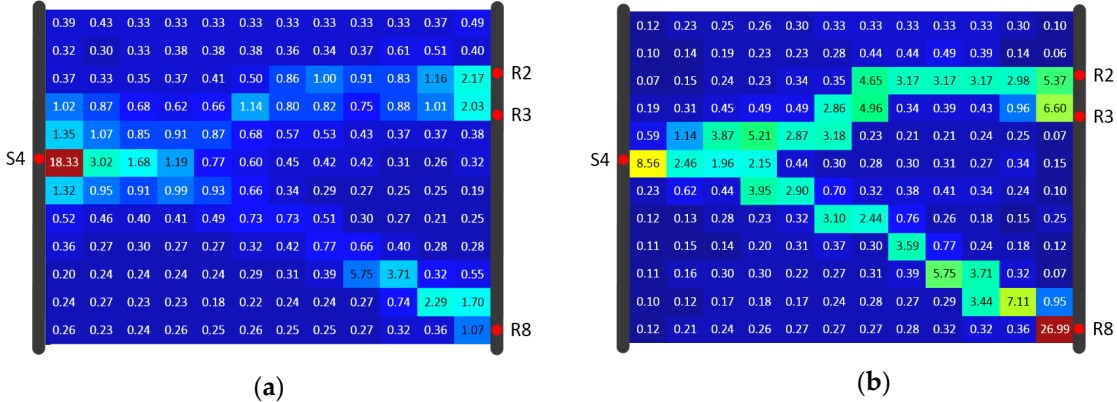

**Figure 19.** Algorithm result comparison for Model B under 12 × 12 resolution of the (**a**) SIRT result and (**b**) SIRT-Cimmino result, shown in diffusivity (m²/s).

**Table 3.** RMSE and correlation coefficient of Model B inversion using SIRT and SIRT-Cimmino.

|  | RMSE | | | Correlation Coefficient | | |
|---|---|---|---|---|---|---|
|  | **8 × 6** | **8 × 8** | **12 × 12** | **8 × 6** | **8 × 8** | **12 × 12** |
| SIRT | 10.39 | 4.66 | 7.11 | 0.64 | 0.63 | 0.61 |
| SIRT-Cimmino | 7.51 | 8.04 | 10.77 | 0.65 | 0.66 | 0.66 |

## 5. Summary and Discussion

In this study, a modified SIRT inversion algorithm with Cimmino iteration is developed for the reconstruction of hydraulic diffusivity distribution within a fluvial aquifer. Two models with different diffusivity distributions are designed to evaluate the performance of this SIRT-Cimmino algorithm. Inversion using a traditional SIRT algorithm is also implemented for comparison purposes.

A residual based result selection rule is proposed and applied to choose the optimal number of iteration steps for the SIRT-Cimmino algorithm. Visual comparison of the reconstructions reveals that the result from the SIRT-Cimmino algorithm reflects the hydraulic features of aquifer better than the conventional SIRT algorithm. This advantage is also proven by comparing the correlation coefficients.

Nevertheless, a deviation between the reconstructed and predefined ("true") diffusivity distribution still exists. This problem is due to the mathematical background, which is necessary for our inversion strategy. The forward modeling employs the groundwater flow equation with parameters $K$ and $S_s$, while the inversion strategy is based on an asymptotic approach, which is directly related to diffusivity $D$ ($D = \frac{K}{S_s}$). Furthermore, we only developed the inversion method with respect to the mathematical algorithm, and did not focus on other inversion aspects, e.g., utilization of prior information, inversion constraints, and inversion strategies. To achieve more potential from this hydraulic travel time based inversion, we suggest the following further investigations:

(a) The overestimation of diffusivity occurs often in model cells near the source and receiver. The reason is the incremental correction built into every iteration step. The diffusivity in a cell is determined by the number of signals traveling through this cell. More signal trajectories in a cell lead to higher estimated diffusivity values, and the cells with a high signal trajectory density are always found at the joints of high diffusivity zones and wells. Depth orientated hydraulic tests (e.g., slug tests) are suggested for parameter estimation in the direct vicinity of the well to provide prior information and inversion constraints. This information could help delimit the inversion values.

(b) Uniform grid setting within the study area affects the accuracy of the ray tracing approximation. Due to the non-uniqueness of the propagation path and the computational burden, the grid is not set to be very fine. An alternative is the adaptive mesh refinement method. With this method, the grid could be refined at the positions where the cells with a higher diffusivity gradient are detected. This method could not only reduce the computational burden but also improve the accuracy of the calculation.

(c) Equal weight for all signals is utilized through the inversion in this study. In reality, an appropriate data subset could deliver better inversion result rather than the whole data set. For example, the inversion results from Hu et al. [14] have shown that pressure signals with a limited source–receiver angle could image horizontal features of the aquifer better. With the help of an appropriate weighting rule, more spatial features could be discovered.

(d) Different types of travel time have different favorites on parameter characterization. Based on the Fermat principle, earlier arrival data reflect the path(s) of higher diffusivity. Hence, inversion results based on early travel times (e.g., $t_{10}$ or $t_{50}$ in Figure 1) could better reconstruct the connectivity of a high diffusivity zone. The late travel times (e.g., $t_{100}$) reflect more integral information of the entire aquifer, so a combination of different kinds of travel times could provide further characterization of aquifer properties.

In future work, we will focus on conditioning the inversion calculation by utilizing prior information, and by introducing earlier signals of travel times to the inversion process.

## Computer Code Availability

The computer code in this work with the name "SIRT-Variants" was developed by Pengxiang Qiu (University of Goettingen, Geoscience Centre, Applied Geology, Goldschmidt Str. 3, 37077, Goettingen, Germany; Telephone number: +49(0)551 39 9267; Email: pqiu@uni-goettingen.de) and Rui Hu (Hohai

University, School of Earth Science and Engineering, Focheng Xi Road 8, 211100, Nanjing, China; Telephone number: +86(0)25 83787174; Email: rhu@hhu.edu.cn). This code is programmed with the language C# (program size < 512KB) and was first available in the year 2018. It functions on a normal computer with standard hardware and the Microsoft Windows operating system. Visual Studio 2013 is required as pre-installed software. This code can be accessed via GitHub through the following link: https://github.com/wichniarek/SIRT-Variants.git.

**Author Contributions:** P.Q. developed the algorithm and wrote the manuscript; R.H. developed the algorithm and wrote the manuscript as a supervisor; L.H. provided necessary modifications to the algorithm and the manuscript; Q.L. designed the numerical modeling; Y.X. performed the inversion work; H.Y. performed the data processing and numerical modeling; J.Q. performed the inversion validation work; T.P. supervised the whole work and provided necessary solutions for various problems.

**Funding:** This work was supported by the Ministry of Education of the People's Republic of China (Project Code 20165037412) and by "the Fundamental Research Funds for the Central Universities" (Project Code 2015B29314). It was also supported by the Jiangsu Provincial Department of Education (Project Code 2016B1203503).

**Acknowledgments:** The first author would like to acknowledge the financial support from the China Scholarship Council (CSC). We also acknowledge the support by the German Research Foundation and the Open Access Publication Funds of the Göttingen University.

**Conflicts of Interest:** The authors declare no conflict of interest.

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
