# Peer review of "A Numerical Study on Travel Time Based Hydraulic Tomography Using the SIRT Algorithm with Cimmino Iteration"

_water, doi:10.3390/w11050909_

Round 1

Reviewer 1 Report

The paper entitled A numerical study on travel time based hydraulic tomography using SIRT algorithm with Cimmino focuses on application of Simultaneous Iterative Reconstruction Technique in Hydraulic Tomography. It is a very interesting topic for pumping tests performance.The advantages of hydraulic travel time based tomography  are high computational efficiency. The article is, however, more methodical due to the simulation of pumping tests. 

Some elements should be improved:

- There is a lack of information about results in the abstract.

-There are mistakes in citations format.

- The description about Dijkstra algorithm should be moved to the previous section.

- It wold be interesting to investigate the travel time type and its influence on model performance (maybe in next paper) and also to perform some pumping tests instead of simulate them.

Author Response

Dear Reviewer,

Thank very much for your comments!

Yes, the advantage of hydraulic tomography is its high efficiency. Not only because of the computational efficiency of the inversion, but also because only the early period (sometime even within seconds) of head data of the pumping tests is needed. Within one day, more than 20 multi well pumping tests could be performed and numerous of drawdown data can be collected. By using our inversion strategy and technique, the result can be obtained even at the same day. However, as we mentioned, there are still several problems related to field application that should be solved. Actually, we are already performing the field test and getting it better. Hopefully the results will be published in our next paper very soon.

1, There is a lack of information about results in the abstract.

Response:

information about results is added (yellow marked), and the abstract is rewritten as following:

Abstract: Travel time based hydraulic tomography is a technique for reconstructing the spatial distribution of aquifer hydraulic properties (e.g. hydraulic diffusivity). Simultaneous Iterative Reconstruction Technique (SIRT) is a widely used algorithm for travel time related inversions. Due to the drawbacks of SIRT implementation in practice, a modified SIRT with Cimmino iteration (SIRT-Cimmino) is proposed in this study. The incremental correction is adjusted, and an iteration-dependent relaxation parameter is introduced. These two modifications enable an appropriate speed of convergence, and the stability of the inversion process. Furthermore, a new result selection rule is suggested to determine the optimal iteration step and its corresponding result. SIRT-Cimmino and SIRT are implemented and verified by using two numerical aquifer models with different predefined diffusivity distributions, where high diffusivity zones are embedded in a homogenous low diffusivity field. Visual comparison of the reconstructions shows that the reconstruction based on SIRT-Cimmino demonstrates the hydraulic features of aquifer better than that from the conventional SIRT algorithm. Root mean square errors and correlation coefficients are also used to quantitatively evaluate the performance of the inversion. The reconstructions based on SIRT-Cimmino are found to preserve the connectivity of the high diffusivity zones and to provide a higher structural similarity to the “true” distribution.

2, There are mistakes in citations format.

Response:

all citations format is now corrected and checked.

3, The description about Dijkstra algorithm should be moved to the previous section.

Response:

thanks, the Dijkstra algorithm description is more related to Methodology, it is now moved to subsection 2.3.

4, It would be interesting to investigate the travel time type and its influence on model performance (maybe in next paper) and also to perform some pumping tests instead of simulating them.

Response:

this is a very good suggestion. Theoretically, the early travel time are mainly related to preferential flow features while the late travel times reflect an integral behavior (this information is provided in section 5, further suggested investigation d). Brauchler et al (2003) compared them in a cylinder rock, and Hu (2011) compared them in field scale pumping test. Their work verified the theory. However, they all used the SIRT algorithm. 

Your suggestion about the performance of field tests is very important. However, this paper focus on the algorithm itself, other influence factors will be considered and investigated in our next paper just like you suggested.

Thanks again for your comments! It made a great improvement for this manuscript!

Best regards,

Pengxiang Qiu

Reviewer 2 Report

With great pleasure, I reviewed the presented manuscript “ A numerical study on travel time based hydraulic tomography using SIRT algorithm with Cimmino iteration.”

In general, I think that the authors did an excellent job by incorporating the Cimmino iterative method with the SIRT algorithm to optimize the estimation accuracy of the diffusivity distribution in two different aquifer models. I recommend it for publication in Water Journal. However, minor revisions need to be addressed:

L121-L139: it seems to me that additional introducing of the t100 data used in the inversion process will be appreciated.

L234-235: (Error! Reference source not 234 found.a), problems of references figures and tables appear many times in the present manuscript (see also in line 238, 243, 265, 287, 288, 290, 324, 326, 336, …)

L248: please unify the unit format; m.s-1 and m-1 or m/s and 1/m and (-) for the porosity.

L428 : I also recommend the using of data with added noisy error to assess the viability of the developed approach in the estimating process

Author Response

Dear Reviewer,

Thank you very much for your comments on our manuscript!

According to your suggestions we have done the following changes:

1, L121-L139: it seems to me that additional introducing of the t100 data used in the inversion process will be appreciated.

Response: please find the attached pdf to read the response, since figures and equations can not be shown properly here. 

2, L234-235: (Error! Reference source not 234 found.a), problems of references figures and tables appear many times in the present manuscript (see also in line 238, 243, 265, 287, 288, 290, 324, 326, 336, …)

Response: thanks, the manuscript showed correctly in my computer, the problem may be on the conversion of files. We checked all the figure and table reference, switch off the automatic cross-reference function, and adjusted them manually.

3, L248: please unify the unit format; m.s-1 and m-1 or m/s and 1/m and (-) for the porosity.

Response: the unit format is unified in “m/s, 1/m and (-)” way.

4, L428: I also recommend the using of data with added noisy error to assess the viability of the developed approach in the estimating process.

Response: this is a very good suggestion. We did consider your suggestion before the submission of this paper. Previous studies show that the noise has little impact on the inversion results if the investigated medium is highly heterogeneous, as the travel time variance is sufficiently large (e.g., Hu et al., 2017).

However, the noise can be more complicated during the field tests. According to field test data from several pumping test that we performed with limited drawdown due to the local aquifer thickness, the drawdown is not large enough to cover the influence of the noise in the early period (less than 20 seconds), and the travel time variance is not sufficiently large. It is really difficult to estimate the travel time. Therefore, we decided only to use computer simulation without field data. Now we have performed pumping tests in another aquifer with larger drawdown with better signal-noise ratio.

Our team considers the noise topic as an important individual topic and did not conclude in theoretical and simulation work. An according paper is under preparation.

Thanks again for your comments! It made a great improvement for this manuscript!

Best regards,

Pengxiang Qiu

Reference:

Hu, L.; Doetsch, J.; Brauchler, R.; Bayer, P. Characterizing CO2 plumes in deep saline formations: Comparison and joint evaluation of time-lapse pressure and seismic tomography. Geophysical 2017, 82.

Water EISSN 2073-4441 Published by MDPI AG, Basel, Switzerland RSS E-Mail Table of Contents Alert
Back to Top